# Psychological Resilience among Left-Behind Children in a Rural Area of Eastern China

**DOI:** 10.3390/children9121899

**Published:** 2022-12-03

**Authors:** Binyan Wang, Lihong Ye, Linshuoshuo Lv, Wei Liu, Fenfen Liu, Yingying Mao

**Affiliations:** 1School of Public Health, Zhejiang Chinese Medical University, Hangzhou 310053, China; 2School of Population Medicine and Public Health, Chinese Academy of Medical Sciences & Peking Union Medical College, Beijing 100730, China

**Keywords:** left-behind children, parent–child communication, psychological resilience, child health

## Abstract

Childhood is an important period for individuals’ psychological development. Due to long-term separation from the parents, left-behind children (LBC) more easily develop deviation in cognition and abnormal personality. In this study, we aimed to explore the status of psychological resilience among LBC in a rural area of eastern China. We carried out a cross-sectional survey including middle and high school students from Qingyuan County of Zhejiang Province. Psychological resilience was measured using a modified scale developed for Chinese children. Data from a total of 1086 participants were collected, and the mean ± standard deviation score of psychological resilience was 4.11 ± 0.42. Multivariable linear regression analyses revealed that being a class leader (*p* = 0.010) and having high self-evaluation of academic performance (*p* < 0.001) were related with psychological resilience. Moreover, high contact frequency between parents and children (*p* = 0.019) was associated with better psychological resilience among LBC. In conclusion, we found that being a class leader and having high self-evaluation of academic performance were associated with better psychological resilience among the children in this rural area and contact between parent and child was an essential factor associated with psychological resilience among LBC.

## 1. Introduction

Since the implementation of the reform and opening-up policy in China in the late 1970s, a great number of laborers from rural areas of China became urban city builders [1]. However, these migrant workers usually do not have equal treatments as citizens in terms of welfare, health care, as well as their children’s accession to education in the cities they migrate to, and most of them are employed in low-paying jobs and living in poor conditions, which further discourages them from bringing their children with them [2]. Therefore, a majority of the migrant workers leave their children at home to be cared for by their relatives, friends, local community, or child institutions to reduce the living costs in the cities. As reported by the National Children’s Fund (UNCIEF), the left-behind children (LBC) refer to the specific group of children who have been left-behind by adult migrants responsible for them, especially one or both parents [3]. The number of the so-called LBC is prominently high in the low- or middle-income countries worldwide. For example, approximately 27 percent of children in Philippine, 36 percent in Ecuador, and more than 40 percent in rural South Africa were estimated to be left behind [4]. In China, there were about 61 million rural children that were left behind by their migrant parents in 2010–2014, and the number has been proliferating [1]. Such labor migration seriously affected the physical and mental health of LBC, who were left uncared for [5].

In recent years, psychological problems of LBC have attracted increasing attention and concerns of the public both at home and abroad. Since long-term separation from their parents, most LBC could not receive adequate care and guidance from their parents and, thus, were at risk for developing psychological and behavioral problems, such as anxiety and depression. For example, it has been reported that the prevalence of psychological and behavioral problems, including loneliness, anxiety, depression, weak interpersonal relationships, and obsessive-compulsive symptoms, was high among LBC [6,7,8,9,10,11,12,13]. Moreover, a cross-sectional study of university students in Jiangsu province in China reported that left-behind experience in childhood was associated with worse mental health of late adolescence [14].

Recent studies related to LBC have placed much emphasis on mental health and its associated factors, while only a few studies have focused on the psychology of mental health from a positive perspective, aiming at investigating its protective factors. Psychological resilience refers to the effective response and adaptation in the face of difficulties or adversity [15]. It is a “rebound ability” with self-protection when life changes pose a challenge or a threat. The historical roots of resilience can be found in two bodies of literature: the psychological aspects of coping and the physiological aspects of stress [16]. Individual-level resilience requires positive experience and good personal relationships and perception of social support to confer resilience. Environmental factors can be imposed on an individual by external resources, such as living environment, school resource, and social assistance [17]. Observational studies have suggested that LBC had lower scores in psychological resilience, which may have more negative symptoms in adverse environments [18,19]. However, most previous studies on LBC’s psychological resilience have not considered the dynamism and complex nature of family relationships, as well as students’ performance at school. For instance, some of LBC’s parents may return to their hometown after migrating for work for an extended period, but previous reports were limited with respect to the differences between current and previous LBC [20]. Furthermore, less attention has been paid to the contact between parents and children, which is an important family process of parent–child communication.

Therefore, in the present study, we aimed to understand the current situation of LBC’s psychological resilience and its associated factors. We conducted a cross-sectional survey of middle school and high school students in Qingyuan County of Zhejiang Province, because Zhejiang Province is a relatively affluent province located in eastern China but there are a great number of LBC in remote mountainous and economically underdeveloped areas, such as Qingyuan county. Moreover, there is no relevant research on psychological resilience of LBC in rural areas of Zhejiang province to date.

## 2. Materials and Methods

### 2.1. Study Participants and Data Collection

The present study was carried out in Qingyuan County in Zhejiang Province, located in eastern China, from August to September in 2017. The protocol of this study was approved by Zhejiang Chinese Medical University Ethical Committee. Stratified random cluster sampling was used to include study participants. Briefly, the middle schools and high schools in Qingyuan county were first stratified into two groups based on their distances to the central town. Then one middle school and one high school were randomly selected from each stratum. All of the students in the selected schools were surveyed using a structured paper questionnaire (*n* = 1086). The survey was performed by School of Public Health at Zhejiang Chinese Medical University in co-operation with local health and educational authorities. We did not provide any incentives to the students.

Socio-demographic characteristics of the participants, including age, sex, ethnicity, location, household income, having siblings or not, type of school, being a class leader or not, self-evaluation of academic performance, as well as left-behind status, were collected using questionnaires by trained investigators. LBC were defined as children under 18 years old who have been left behind at their original residence for at least six months while one or both parents migrated to cities for work and were classified into “father migrated”, “mother migrated”, “both migrated”, “past migrated”, and a reference group of children (non-LBC). For LBC, parental migration status, type of caregivers, contact frequency between parent(s) and child, and frequency of parent(s) visiting the child were further collected.

The study participants’ psychological resilience was assessed using a modified scale compiled by Hu et al. [21] from the Department of Psychology at Peking University in 2008, which included 27 items from two dimensions, personal strength and support. Personal strength included three factors: goal focus, emotion control, and positive cognition, and support included two factors: family support and interpersonal assistance. Participants were asked to indicate the extent to which they agreed with each item on a 5-point Likert rating scale (1 = strongly disagree; 5 = strongly agree). The scale was validated among children in China, and the internal consistency of the initial survey (*n* = 283) was 0.85 and the reliability of the retest (*n* = 420) was 0.83 [21].

### 2.2. Quality Control

The questionnaire used in the present study was developed after pre-survey and revision, and the investigators were trained before the survey. Before the survey, all participants were informed of the purpose of this study and the guidelines of the questionnaire. Participants were encouraged to answer the questions truthfully. The questionnaires were collected on site within a specified time. After the survey, on the same day, the investigators checked the quality of the questionnaire. If problems were found, a timely return visit was made. EpiData 3.02 software (EpiData Association, Denmark) was used to input the questionnaire data independently by two investigators, and the database was established after cross-validation.

### 2.3. Statistical Analysis

Statistical analysis was performed using R 3.3.0 (R core team). The scores of each item of psychological resilience were expressed in mean ± standard deviation (SD). The differences between groups were compared using two-sample t-test and one-way analyses of variance (ANOVA) for continuous variables, and chi-square test for categorical variables. Associated factors for psychological resilience were further analyzed using multivariable linear regression models. Two-sided *p*-values less than 0.05 were considered statistically significant.

## 3. Results

### 3.1. The Basic Characteristics of Study Participants

This study included a total of 1086 children aged 12–18 years from middle and high schools (Table 1). The percentage of females was 54.42%, and 97.42% of them were of Han ethnicity. Among them, 365 (33.6%) children were left behind, with 162 (44.4%) classified as both parents migrated, 142 (38.9%) as father migrated, 14 (3.8%) as mother migrated, and 47 (12.9%) as past migrated. In terms of caregivers, 162 (44.4%) were cared by single parent, 123 (33.7%) by grandparents, 49 (13.4%) by uncles and aunts, 12 (3.3%) by brothers/sisters, and 19 (5.2%) by themselves.

Compared with non-LBC, there were more LBC with parents who were divorced or passed away (*p* = 0.002). There were no differences in the distributions of gender, ethnicity, household income, having sibling(s), being a class leader, and self-evaluation of academic performances were found between LBC and non-LBC (*p* > 0.05).

### 3.2. The Psychological Resilience and Its Associated Factors among the Study Participants

As shown in Table 1, the average score of psychological resilience was higher in students from the town than those from the village (*p* = 0.019), and in class leaders than non-class leaders (*p* < 0.001). Higher household income and high self-evaluation of academic performances were associated with higher scores of psychological resilience, respectively (*p* = 0.003 and *p* < 0.001).

For the dimension of personal strength, the average score was higher in boys (*p* = 0.025), middle school students (*p* < 0.001), class leaders (*p* < 0.001), and those with higher household income (*p* < 0.001) and better self-evaluation of academic performances (*p* < 0.001). For the dimension of support, girls (*p* < 0.001), high school students (*p* < 0.001), those living in the town (*p* = 0.005), being a class leader (*p* = 0.039), and having better self-evaluation of academic performances scored better (*p* = 0.006).

In the multivariable linear regression models, statistically significant associations were observed for being a class leader (β = 0.067, *p* = 0.010) and better self-evaluation of academic performances (β = 0.074, *p* < 0.001) with higher scores of psychological resilience. For its two dimensions, higher household income (β = 0.083, *p* = 0.032), being a class leader (β = 0.085, *p* = 0.017) and better self-evaluation of academic performances (β = 0.110, *p* < 0.001) were associated with higher scores of personal strengths, while living in a village or town (β = 0.078, *p* = 0.006) and better self-evaluation of academic performances (β = 0.029, *p* = 0.013) were related with higher scores in support (Table 2).

### 3.3. The Psychological Resilience and Associated Factors in Left-Behind Children

However, the univariable and multivariable analysis did not show statistically significant differences in psychological resilience between LBC and non-LBC, nor in its two dimensions. Among LBC, in the univariable analysis, being a class leader (*p* = 0.006), higher self-evaluation of academic performances (*p* < 0.001), frequent contact between parent and child (*p* = 0.004), and frequent parental visit to children (*p* = 0.028) were associated with higher scores of psychological resilience (Table 3). Type of school (*p* = 0.009), being a class leader (*p* = 0.029), and higher self-evaluation of academic performances (*p* < 0.001) were associated with higher scores of personal strength, while sex (*p* = 0.027), high school students (*p* = 0.001), being a class leader (*p* = 0.017), and higher self-evaluation of academic performances (*p* = 0.003) were associated with higher scores of support. Moreover, psychological resilience among LBC was associated with the contact frequency between parents and children, with those contacting daily scoring the highest (4.20 ± 0.46) and those contacting once per 30 days or more scoring the lowest (3.96 ± 0.48) (*p* = 0.004). Similar scores and trends were found for both personal strength (*p* = 0.019) and support (*p* = 0.012).

As shown in the Table 4 and Table 5, in the multivariable regression analysis, being a class leader (β = 0.121, *p* = 0.010 in Table 4; β = 0.106, *p* = 0.023 in Table 5) and higher self-evaluation of academic performance (β = 0.100 in Table 4, β = 0.108 in Table 5, both *p* < 0.001) and high contact frequency between parents and children (β = 0.041, *p* = 0.019 in Table 4) were associated with psychological resilience. For its two dimensions, higher self-evaluation of academic performance (β = 0.140 in Table 4, β = 0.147 in Table 5, both *p* < 0.001) was associated with personal strength, and being a class leader (β = 0.116, *p* = 0.020 in Table 4, β = 0.101, *p* = 0.043 in Table 5), self-evaluation of academic performance (β = 0.049, *p =* 0.019 in Table 4, β = 0.058, *p* = 0.005 in Table 5), and contact frequency between parents and children (β = 0.045, *p* = 0.017 in Table 4) were associated with support.

Among non-LBC, univariable linear regression analysis showed that living in the town area (*p* = 0.033), higher household income (*p* = 0.012), being a class leader (*p* = 0.023), and self-evaluation of academic performance (*p* < 0.001) were associated with higher scores of psychological resilience (Table 6). Sex (*p* = 0.033), higher household income (*p* = 0.007), type of school (*p* = 0.009), being a class leader (*p* = 0.013) and higher self-evaluation of academic performances (*p* < 0.001) were associated with higher scores of personal strength, while sex (*p* < 0.001), living in the town (*p* = 0.006), and type of school (*p* = 0.001) were associated with higher scores of support. However, we did not observe the potential association between being a class leader (*p* = 0.420), self-evaluation of academic performance (*p* = 0.269) and support. In the multivariable linear regression analysis, being a class leader (β = 0.106, *p* = 0.024) and higher self-evaluation of academic performance (β = 0.106, *p* < 0.001) were associated with psychological resilience. In its two dimensions, higher self-evaluation of academic performance (β = 0.144, *p* < 0.001) was associated with personal strength. Being a class leader (β = 0.100, *p* = 0.045) and higher self-evaluation of academic performance (β = 0.058, *p* = 0.005) were associated with support (Table 7).

## 4. Discussion

To the best of our knowledge, this is the first study to examine psychological resilience of children living in the rural areas of Zhejiang province in eastern China. In the present study, we found that being a class leader and self-evaluation of academic performance were associated with psychological resilience in this rural area of eastern China, and parent–child contact is an essential factor for psychological resilience among LBC.

There are several theories of resilience: (i) resilience is a positive psychological outcome among high-risk individuals [22]; (ii) resilience is a dynamic, interactive process that involves stress, pressure, and other adverse life events [23]; and (iii) resilience is the ability of an individual to cope with stress, frustration, trauma, and other adverse life events [24]. Compared to the first and the second theories, the theory of resilience as ability is largely measurable and is most directly amenable to therapeutic interventions. In the past few decades, studies have suggested the negative impact of parental migration on children’s psychological resilience [19]. However, in the present study, we did not observe differences in scores of psychological resilience between LBC and non-LBC, which was not similar with results from previous studies [25]. This may be because the study participants in our study from Qingyuan County encountered various environmental diversities and LBC accounted for only one third of the study population. Moreover, studies showed that economic conditions were closely related to children’s physical and mental health [26,27], since Zhejiang province is relatively affluent, the average levels of psychological resilience of children in the present study are relatively high. The students in this special area may have similar coping capacities with their general rural adversities or socio-economic challenges, despite their parental migration status. It is also possible that, due to the limited sample size, this study did not have enough statistical power. However, the results from our study are in consistency with the findings of another cross-sectional survey conducted in Yunyang County of Three Gorges Areas in the middle of China, which reported no differences in the resilience between LBC and non-LBC in middle school students [28].

As for the factors associated with psychological resilience among children living in the mountainous areas, we found that being class leader and self-evaluation of academic performances were potential determinants of psychological resilience. Results from the study by Wang et al. also showed that LBC with learning disabilities had statistically significantly lower scores of psychological resilience than those with excellent and median learning abilities [29]. Because children have learning pressure at school, their academic performance directly determines their psychological toughness. Therefore, our study suggested that teachers and parents should actively encourage students to help improve their academic performances. Moreover, our study also suggested that the scores of psychological resilience of the class leader students were higher than those of non-class leaders. The possible reasons may be that students who serve as class leaders have higher self-social value, and their self-esteem is easier to be satisfied, which is positively correlated with children’s mental health. Therefore, it is suggested that schools pay more attention to cultivating the students’ self-social value and self-esteem, especially for LBC.

Moreover, we also found that among LBC, parental–child contact was associated with psychological resilience. It is in consistency with findings from Zhou et al. that better parent–child communication was associated with better development of their mental health among LBC [19]. Results from the study by Wang et al. also showed that contact frequency with parents was closely related to LBC’s mental health [30]. It could be seen that maintaining a high contact frequency was very important for the parents of migrant workers to timely and comprehensively grasp the living and school life of their children and maintain an intimate parent–child relationship. In addition, as the study conducted by Clarke et al. suggested, children who suffered from insecure parent relationships were more difficult in building resilience [31]. Therefore, communication between parents and children is necessary for the development of psychological resilience of LBC, leading to our suggestions that the parents of LBC should communicate with their children as much as possible and pick up their children for a reunion as much as possible. The study by Liu et al. evaluated the impact of parental migration and parent–child relation types on psychological resilience of LBC and found that mother’s remote migration had a significantly negative impact on psychological resilience of LBC [18]. In this study, though we did not detect statistically significant differences in psychological resilience among LBC with different migration statuses, our results showed that the scores of psychological resilience of LBC with mother out for work was the lowest. A study conducted in Ghana and Nigeria found similar results [32]. Since migrant mothers cannot take good care of their children’s daily life, they also have difficulty in giving appropriate care and support emotionally; it is suggested that mothers stay at home to take care of LBC as much as possible.

This study has several limitations. First, this survey only included middle and high school students in Qingyuan, a rural area of Zhejiang province in eastern China. Further studies should broaden the research objectives and include students from other grades and from nonrural areas. Second, the data in the present study were collected by self-reported questionnaires with limited variables from the children, so further studies are warranted to collect further information on mental health and trauma exposure, which may also influence psychological resilience. In addition, this is a cross-sectional study with limited capacity in causal inference; therefore, we could not rule out the possibility that psychological resilience might contribute to children’s academic performance or their contact with their parents. Further longitudinal studies are needed to investigate the causal associations between identified factors and psychological resilience and its potential underlying mechanisms.

## 5. Conclusions

In summary, this study suggested that being a class leader and self-evaluation of academic performance were related with psychological resilience in children in this rural area of eastern China, and parent–child contact is an important factor for psychological resilience among LBC. Targeted intervention programs, such as strengthening parental–child communications should be delivered to improve the psychological resilience among LBC in China.

## Figures and Tables

**Table 1 children-09-01899-t001:** Psychological resilience among the study participants.

Variables		*n*	%	Psychological Resilience			Personal Strength			Support		
				M	SD	*p*	M	SD	*p*	M	SD	*p*
Sex	Female *	591	54.42	4.12	0.42	0.396	4.02	0.57	**0.025**	4.25	0.42	**<0.001**
	Male	495	45.58	4.10	0.43		4.10	0.59		4.10	0.43	
Ethnicity	Han *	1058	97.42	4.11	0.42	0.870	4.05	0.58	0.944	4.18	0.44	0.553
	Others	28	2.58	4.12	0.46		4.05	0.65		4.22	0.38	
Location	Village *	346	31.86	4.07	0.41	**0.019**	4.02	0.54	0.171	4.13	0.44	**0.005**
	Town	740	68.14	4.13	0.43		4.07	0.60		4.21	0.43	
^1^ Type of school	Middle school * student	479	44.11	4.12	0.44	0.403	4.14	0.59	**<0.001**	4.11	0.44	**<0.001**
	High school student	607	55.89	4.10	0.41		3.99	0.56		4.24	0.42	
Having sibling(s)	No *	830	71.14	4.12	0.41	0.458	4.06	0.57	0.816	4.19	0.43	0.194
	Yes	256	23.57	4.09	0.45		4.05	0.62		4.15	0.44	
Left-behind	No *	721	66.39	4.11	0.41	0.652	4.05	0.58	0.959	4.19	0.43	0.360
	Yes	365	33.61	4.10	0.44		4.05	0.59		4.16	0.45	
^2^ Household income	Low *	188	17.31	4.03	0.45	**0.003**	3.96	0.60	**<0.001**	4.12	0.47	0.361
	Moderate	849	78.18	4.12	0.41		4.06	0.56		4.20	0.42	
	High	49	4.51	4.18	0.54		4.27	0.70		4.07	0.45	
^3^ Class leader	No *	685	63.08	4.08	0.42	**<0.001**	4.01	0.57	**<0.001**	4.16	0.44	**0.039**
	Yes	401	36.92	4.17	0.43		4.13	0.59		4.22	0.43	
Self-evaluation of academic performances	Bad *	198	18.23	3.94	0.44	**<0.001**	3.80	0.56	**<0.001**	4.12	0.48	**0.006**
	Below average	174	16.02	4.05	0.39		3.97	1.52		4.15	0.44	
	Average	386	35.54	4.14	0.38		4.11	0.54		4.19	0.41	
	Above average	270	24.86	4.20	0.42		4.18	0.59		4.22	0.42	
	Good	58	5.34	4.25	0.54		4.25	0.72		4.25	0.50	

^1^ Students in Grade 3 to 6 and Grade 7 to 8 were classified into middle school students and high school students, respectively. ^2^ The household income was self-rated as low, moderate, or high by the study participants. ^3^ The class leader was the students who help teachers to manage the class affairs. * Reference group. The bold indicated *p* < 0.05. Abbreviations: M, mean; SD, standard deviation.

**Table 2 children-09-01899-t002:** Multivariable linear regression analysis of factors associated with psychological resilience.

Variables	Psychological Resilience				Personal Strength				Support			
	β	se	t	*p*	β	se	t	*p*	β	se	t	*p*
Location	0.052	0.027	1.927	0.054	0.032	0.037	0.864	0.388	0.078	0.029	2.729	**0.006**
^1^ Household income	0.047	0.028	1.663	0.097	0.083	0.039	2.148	**0.032**	0.002	0.030	0.056	0.956
^2^ Class leader	0.067	0.026	2.584	**0.010**	0.085	0.036	2.399	**0.017**	0.045	0.027	1.641	0.101
Self-evaluation of academic performances	0.074	0.011	6.706	**<0.001**	0.110	0.015	7.317	**<0.001**	0.029	0.012	2.494	**0.013**

^1^ The household income was self-rated as low, moderate, or high by the study participants. ^2^ The class leader was the students who help teachers to manage the class affairs. The variables that reach the statistically significant association for psychological resilience (*p* < 0.05) in the univariable analysis were included in the multivariable linear regression models. The bold indicated *p* < 0.05. Abbreviations: se, standard error.

**Table 3 children-09-01899-t003:** Psychological resilience among left-behind children.

Variables		No	%	Psychological Resilience			Personal Strength			Support		
				M	SD	*p*	M	SD	*p*	M	SD	*p*
Sex	Female *	179	49.04	4.11	0.43	0.731	4.03	0.57	0.373	4.22	0.46	**0.027**
	Male	186	50.96	4.09	0.46		4.08	0.61		4.11	0.44	
Ethnicity	Han *	358	98.08	4.10	0.44	0.546	4.05	0.59	0.867	4.17	0.46	0.129
	Others	7	1.92	3.99	0.47		4.01	0.66		3.96	0.30	
Location	Village *	123	33.70	4.07	0.43	0.297	4.02	0.55	0.438	4.13	0.45	0.306
	Town	242	66.30	4.12	0.45		4.07	0.61		4.18	0.46	
^1^ Type of school	Middle school *	181	49.59	4.11	0.45	0.651	4.13	0.61	**0.009**	4.09	0.45	**0.001**
	High school	184	50.41	4.09	0.43		3.97	0.56		4.24	0.45	
Having sibling(s)	No *	92	25.21	4.12	0.43	0.115	4.07	0.58	0.267	4.19	0.45	0.081
	Yes	273	74.79	4.04	0.47		3.99	0.61		4.09	0.45	
^2^ Household income	Low *	69	18.90	4.03	0.47	0.109	3.97	0.60	0.052	4.10	0.47	0.716
	Moderate	275	75.34	4.12	0.43		4.06	0.57		4.19	0.45	
	High	21	5.75	4.16	0.54		4.27	0.74		4.02	0.43	
^3^ Class leader	No *	127	34.79	4.06	0.45	**0.006**	4.00	0.58	**0.029**	4.12	0.47	**0.017**
	Yes	238	65.21	4.19	0.42		4.15	0.60		4.24	0.42	
Self-evaluation of academic performances	Bad *	74	20.27	3.91	0.48	**<0.001**	3.77	0.58	**<0.001**	4.08	0.50	**0.003**
	Below average	55	15.07	3.94	0.45		3.86	0.53		4.05	0.50	
	Average	126	34.52	4.16	0.38		4.14	0.54		4.19	0.41	
	Above average	94	25.75	4.24	0.41		4.25	0.59		4.22	0.40	
	Good	16	4.38	4.26	0.50		4.18	0.60		4.36	0.56	
Parental migration status	Father-migrated *	142	38.90	4.08	0.43	0.291	4.05	0.56	0.610	4.13	0.46	0.131
	Mother-migrated	14	3.84	3.98	0.52		3.84	0.60		4.15	0.56	
	Both-migrated	162	44.38	4.11	0.44		4.05	0.59		4.18	0.46	
	Past migrated	47	12.88	4.17	0.47		4.12	0.67		4.24	0.40	
Type of caregivers	Single-parent *	162	44.38	4.13	0.42	0.413	4.10	0.54	0.259	4.16	0.45	0.961
	Grandparents	123	33.70	4.05	0.45		4.00	0.59		4.12	0.46	
	Uncles/aunts	49	13.42	4.19	0.42		4.09	0.61		4.31	0.41	
	Brothers/sisters	12	3.29	4.17	0.46		4.06	0.68		4.31	0.43	
	By oneself	19	5.21	3.94	0.56		3.91	0.81		3.98	0.52	
Contact frequency with parents	1 per 30 days or more *	43	11.78	3.96	0.48	**0.004**	3.98	0.63	**0.019**	3.94	0.50	**0.012**
	1 per 15–30 days	71	19.45	4.07	0.38		3.97	0.50		4.19	0.38	
	1 per 4–7days	107	29.32	4.09	0.45		4.02	0.60		4.17	0.46	
	1 per 2–3 days	69	18.90	4.14	0.43		4.09	0.58		4.21	0.47	
	Daily	75	20.55	4.20	0.46		4.18	0.62		4.22	0.43	
Frequency of parents visiting children	More than 1 year *	32	8.77	4.05	0.48	**0.028**	4.04	0.68	0.052	4.07	0.52	0.113
	Every 6 months to 1 year	69	18.90	4.09	0.46		4.01	0.60		4.19	0.46	
	Every 1–6 months	153	41.92	4.05	0.40		3.94	0.54		4.12	0.43	
	Every month	111	30.41	4.20	0.46		4.18	0.61		4.23	0.45	

^1^ Students in Grade 3 to 6 and Grade 7 to 8 were classified into middle school students and high school students, respectively. ^2^ The household income was self-rated as low, moderate, or high by the study participants. ^3^ The class leader was the students who help teachers to manage the class affairs. * Reference group. The bold indicated *p* < 0.05. Abbreviations: M, mean; SD, standard deviation.

**Table 4 children-09-01899-t004:** Multivariable linear regression analysis of factors associated with psychological resilience among left-behind children.

Variables	Psychological Resilience				Personal Strength				Support			
	β	se	t	*p*	β	se	t	*p*	β	se	t	*p*
^1^ Class leader	0.121	0.047	2.585	**0.010**	0.114	0.062	1.842	0.066	0.116	0.050	2.333	**0.020**
Self-evaluation of academic performances	0.100	0.019	5.131	**<0.001**	0.140	0.026	5.472	**<0.001**	0.049	0.021	2.364	**0.019**
Contact frequency with parents	0.041	0.018	2.348	**0.019**	0.034	0.023	1.481	0.140	0.045	0.019	2.403	**0.017**

^1^ The class leader was the students who help teachers to manage the class affairs. The variables that reach the statistically significant association for psychological resilience (*p* < 0.05) in the univariable analysis were included in the multivariable linear regression models. To avoid collinearity, “Contact frequency with parents” and “Frequency of parents visiting children” were added to the linear regression model, respectively. The bold indicated *p* < 0.05. Abbreviations: se, standard error.

**Table 5 children-09-01899-t005:** Multivariable linear regression analysis of factors associated with psychological resilience among left-behind children.

Variables	Psychological Resilience				Personal Strength				Support			
	β	se	t	*p*	β	se	t	*p*	β	Se	t	*p*
^1^ Class leader	0.106	0.047	2.278	**0.023**	0.101	0.061	1.642	0.101	0.101	0.050	2.028	**0.043**
Self-evaluation of academic performances	0.108	0.019	5.631	**<0.001**	0.147	0.025	5.833	**<0.001**	0.058	0.020	2.832	**0.005**
Frequency of parents visiting children	0.047	0.024	1.954	0.051	0.054	0.032	1.711	0.088	0.037	0.026	1.427	0.154

^1^ The class leader was the students who help teachers to manage the class affairs. The variables that reach the statistically significant association for psychological resilience (*p* < 0.05) in the univariable analysis were included in the multivariable linear regression models. To avoid collinearity, “Contact frequency with parents” and “Frequency of parents visiting children” were added to the linear regression model, respectively. The bold indicated *p* < 0.05. Abbreviations: se, standard error.

**Table 6 children-09-01899-t006:** Psychological resilience among non-left-behind children.

Variables		No	%	Psychological Resilience			Personal Strength			Support		
				M	SD	*p*	M	SD	*p*	M	SD	*p*
Sex	Female *	405	37.29	4.13	0.41	0.436	4.01	0.570	**0.033**	4.26	0.407	**<0.001**
	Male	316	29.10	4.10	0.42		4.11	0.58		4.09	0.43	
Ethnicity	Han *	700	64.46	4.11	0.41	0.583	4.05	0.57	0.986	4.19	0.43	0.147
	Others	21	1.93	4.17	0.46		4.06	0.66		4.31	0.66	
Location	Village *	223	20.53	4.07	0.40	**0.033**	4.02	0.43	0.261	4.12	0.43	**0.006**
	Town	498	45.86	4.14	0.42		4.07	0.59		4.22	0.42	
^1^ Type of school	Middle school *	181	49.59	4.13	0.43	0.451	4.14	0.58	**0.009**	4.12	0.44	**0.001**
	High school	184	50.41	4.10	0.40		4.00	0.56		4.24	0.41	
Having sibling(s)	No *	557	51.29	4.11	0.40	0.757	4.05	0.56	0.595	4.19	0.43	0.809
	Yes	164	15.10	4.12	0.44		4.08	4.08		4.18	0.42	
^2^ Household income	Low *	69	18.90	4.03	0.60	**0.012**	3.95	3.95	**0.007**	3.95	0.47	0.382
	Moderate	275	75.34	4.24	0.56		0.77	0.77		4.20	0.41	
	High	21	5.75	4.19	0.55		4.26	0.69		4.11	0.47	
^3^ Class leader	No *	447	41.16	4.09	0.40	**0.023**	4.01	0.57	**0.013**	4.18	0.42	0.420
	Yes	274	25.23	4.16	0.43		4.12	0.58		4.21	0.43	
Self-evaluation of academic performances	Bad *	124	11.42	3.96	0.42	**<0.001**	3.82	0.55	**<0.001**	4.15	0.47	0.269
	Below average	119	10.96	4.10	0.36		4.03	0.50		4.20	0.40	
	Average	260	23.94	4.13	0.38		4.09	0.54		4.19	0.41	
	Above average	176	16.21	4.17	0.43		4.14	0.59		4.22	0.43	
	Good	42	3.87	4.24	0.56		4.27	0.77		4.20	0.47	

^1^ Students in Grade 3 to 6 and Grade 7 to 8 were classified into middle school students and high school students, respectively. ^2^ The household income was self-rated as low, moderate, or high by the study participants. ^3^ The class leader was the students who help teachers to manage the class affairs. * Reference group. The bold indicated *p* < 0.05. Abbreviations: M, mean; SD, standard deviation.

**Table 7 children-09-01899-t007:** Multivariable linear regression analysis of factors associated with psychological resilience among non-left-behind children.

Variables	Psychological Resilience				Personal Strength				Support			
	β	se	t	*p*	β	se	t	*p*	β	se	t	*p*
Location	0.037	0.048	0.783	0.434	0.028	0.064	0.438	0.662	0.048	0.050	0.954	0.341
^1^ Household income	0.028	0.048	0.590	0.556	0.062	0.064	0.966	0.335	−0.013	0.051	−0.260	0.795
^2^ Class leader	0.106	0.047	2.272	**0.024**	0.112	0.060	1.8	0.073	0.100	0.049	2.014	**0.045**
Self-evaluation of academic performances	0.106	0.019	5.445	**<0.001**	0.144	0.026	5.562	**<0.001**	0.058	0.021	2.820	**0.005**

^1^ The household income was self-rated as low, moderate, or high by the study participants. ^2^ The class leader was the students who help teachers to manage the class affairs. The variables that reach the statistically significant association for psychological resilience (*p* < 0.05) in the univariable analysis were included in the multivariable linear regression models. The bold indicated *p* < 0.05. Abbreviations: se, standard error.

## Data Availability

The datasets analyzed in this study are available from the corresponding author on reasonable request.

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
