# Peer review of "Psychological Resilience among Left-Behind Children in a Rural Area of Eastern China"

_children, 2022, doi:10.3390/children9121899_

Round 1

Reviewer 1 Report

Introduction

1.       Page 1- line 39- so-call should be so-called

2.       Page 1 -With regards to the statement on the number of children left behind by parents, are all 61 million a result of migrant workers?

3.       Page 2 lines 46-29 require citations to support the statements on care/ guidance, mental health and personality outcomes. Lines 50-52 could be combined with lines 46-49 to reduce repetition.

4.       Page 2- I think a cleaner transition would be to start the paragraph on resilience why it is also important to study protective factors, followed by the definition of resilience and the statistics on LBC, or lack thereof. Citations are needed for any studies on psychological resilience (lines 69-74).

5.       This is an interesting study to conduct. I think to make the introduction stronger for readership, another paragraph supporting the implications of resilience on parent-child processes, and long term outcomes would support the rationale for the study.

6.       Page 2- lines 76-83. Perhaps the authors can start with stating the goal of the study, and then explain why they chose that County, rather than the reverse.

Methods

1.       Can the authors elaborate on how students were contacted and how the survey was delivered (pen/paper, electronic).

2.       Were incentives provided? Were the surveys done at school?

3.       What were the eligibility criteria? Were the authors specifically targeting LBC or all children?  

4.       How many schools were in the strata and in the final selection?

5.       In terms of measures, were other demographic data collected, such as demographics on year of education, age, sex,etc.?

6.       Page 2, line 115- The sentence beginning with “during the survey” reads a bit awkwardly. Maybe the authors can rephase to state that guidelines on the study purpose and questionnaire were provided to participants.

7.       The authors describe gender and other items measured in the results section. I would note that demographics and academic performance/ leadership were also part of the measures earlier in the methods.

8.       What were the reference groups for each variable?

9.       Within the tables, I think it may be sufficient to say p< .001 when the values are exponentially small.

10.   In the methods, the authors mention that psychological reliance was comprised of two dimensions- strength and support. I find it interesting that the individual domains were significant but not the composite scores across a number of variables (school type, sex). Can the authors comment on this?

11.   While frequency of parents visiting children was not associated with psychological resilience, it could be that the current breakdown of categories may influence this trending association. Perhaps the authors could consider collapsing 1-3 and 3-6months as 1-6 months, to be consistent with  the 6 months- 1 year category.

12.   If family support is a dimension of support for psychological resistance, would this potentially indicate collinearity with the author’s variable for frequency of parent contact?

Additional considerations

1.       Given the focus of academic and school indicators in the study, I think it would be helpful to connection academic functioning with resilience in the introduction. As it stands, the introduction is more heavily focused on family processes.

2.       Looking at the individual dimensions of the psychological resilience scale, it is also plausible that psychological resilience, parental contact, etc, may contribute to child success in academic settings rather than having psychological resilience act as the outcome. These should be considered in the limitations or discussion if data are not available to the authors for their analyses.

3.       Other variables such as mental health, trauma exposure, emotion regulation may all influence psychological resilience.

4.       Page 16. The authors discuss that better parental-child communication was associated with better resilience. I do not see these data in the results, unless this the same as contact frequency? Perhaps they may define contact frequency in the methodology.

Reviewer 2 Report

This is a good piece of work focusing on mental health in left-behind children. Although the main messages are conveyed, I believe that many aspects need to be improved. My overall suggestions are (1) checking and improving the grammar, vocabularies, formats of tables and writing style; (2) having your paper professionally edited by an English-language expert and/or have a thorough review of the English presentation to improve the clarity and readability of the whole manuscript

Comment 1: Line 30-31, […a large number of laborers from rural areas of China have joined as urban city builders], please add reference(s) for this sentence.

Comment 2: Line 36-39, [As reported by the National Children’s Fund (UNCIEF), the left-behind children (LBC) refer to the specific group of children who have been left behind by adult migrants responsible for them, especially one or both parents], please cite the report as a reference.

Comment 3: Line 47-48, [Therefore it is easier for them to develop deviation in cognition and abnormal personality, and their mental health status is generally worse than that of non-LBC], any evidence to support this statement?

Comment 4: Line 70-73, [For instance, some of the LBC’s parents may return to their hometown after migrating for work for an extended period, but previously collected data were limited with respect to the differences between current-LBC and previous-LBC], please add reference(s) for this example.

Comment 5: What is the rationale of this study? Add previous reviews on psychological resilience and associated factors among LBS in introduction.

Comment 6: In [2.1. Study participants and data collection], please add sample numbers.

Comment 7: What are inclusion and exclusion criteria for sample selection?

Comment 8: Were there missing data in this study? If the answer was yes, please add reasons for missing data

Comment 9: Please make the descriptions of results section more concise. The numbers can be found in the tables, and please try not to repeat them in the main text again unless they are especially important. The comment applies to all paragraphs in results section.

Comment 10: Add and explain more for the discussion on associated factors with psychological resilience.

Round 2

Reviewer 1 Report

I thank the authors for responding and addressing my concerns with the manuscript entitled "Psychological resilience among left-behind children in a rural area of Eastern China.

I have just a few suggestions for additional clarity in the manuscript and would recommend one additional proofreading.

Introduction:

The authors have changed ‘their parents’ to ‘the parents’- lines 56/57.  I would recommend using ‘their’.

Change ‘could not get’ to ‘could not receive’ - line 57

Change laid to ‘placed’- line 67

Change previously to previous- line 88

Methods.

I appreciate the authors' responses to my questions on clarifying the procedures around the survey administration. Based on their response, a sentence or two should be included on incentives and this extra piece to the recruitment strategy, which summarizes their response to my inquiry. For reference, here is a truncated piece of the response that I make reference  to: We did not provide any incentives to the students. Because this study was funded by the Department of Civil Affairs of Zhejiang province, the local education administration helped us to contact the schools and teachers. Then we have trained investigators went into the classrooms with the teachers from the schools to facilitate the survey.

Reviewer 2 Report

The authors have addressed my commnets

Author Response

Thank you for reviewing our manuscript.